# How Antibodies Recognize Pathogenic Viruses: Structural Correlates of Antibody Neutralization of HIV-1, SARS-CoV-2, and Zika

**DOI:** 10.3390/v13102106

**Published:** 2021-10-19

**Authors:** Morgan E. Abernathy, Kim-Marie A. Dam, Shannon R. Esswein, Claudia A. Jette, Pamela J. Bjorkman

**Affiliations:** 1Division of Biology and Biological Engineering, California Institute of Technology, Pasadena, CA 91125, USA; mabernat@caltech.edu (M.E.A.); kdam@caltech.edu (K.-M.A.D.); cajette@caltech.edu (C.A.J.); 2David Geffen School of Medicine at University of California, Los Angeles, CA 90095, USA; sesswein@caltech.edu

**Keywords:** antibody, cryo-electron microscopy, HIV-1, SARS-CoV-2, structural biology, virus, X-ray crystallography, Zika

## Abstract

The H1N1 pandemic of 2009-2010, MERS epidemic of 2012, Ebola epidemics of 2013-2016 and 2018-2020, Zika epidemic of 2015-2016, and COVID-19 pandemic of 2019-2021, are recent examples in the long history of epidemics that demonstrate the enormous global impact of viral infection. The rapid development of safe and effective vaccines and therapeutics has proven vital to reducing morbidity and mortality from newly emerging viruses. Structural biology methods can be used to determine how antibodies elicited during infection or vaccination target viral proteins and identify viral epitopes that correlate with potent neutralization. Here we review how structural and molecular biology approaches have contributed to our understanding of antibody recognition of pathogenic viruses, specifically HIV-1, SARS-CoV-2, and Zika. Determining structural correlates of neutralization of viruses has guided the design of vaccines, monoclonal antibodies, and small molecule inhibitors in response to the global threat of viral epidemics.

## 1. Introduction

Advances in structural biology in recent decades have played a key role in the determination of disease-relevant protein complexes and guided the design of new therapeutics and vaccines. An early pioneer in structural biology was the X-ray crystallographer Rosalind Franklin. While she is best known for her role in collecting the X-ray fiber diffraction patterns that revealed the 3D structure of DNA, her contributions in biologically-related fields also included insights into the structures of protein encapsulated viruses such as tobacco mosaic virus (TMV), poliovirus, and turnip yellow mosaic virus. During Franklin’s studies of viruses in the 1950s, a central question was how viruses managed to build a protein shell to shield their genetic material given that only a limited number of viral capsid proteins could be encoded within a viral genome based on capsid size constraints. Franklin’s X-ray analysis revealed the arrangement of the protein subunits in TMV, allowing her to create the first three-dimensional model of a virus [1,2,3,4]. Following this work, she used X-ray data to determine the position and orientation of RNA packaged inside of the rod-shaped TMV [5]. Unlike prior speculation that placed the RNA at the center of the rod, her work revealed the virus was hollow, which led to the discovery that the RNA spiraled with the helical protein capsid. This work was fundamental in understanding principles of virus structure. Franklin’s contributions to the field of virology are summarized on her tombstone, which reads, “Her research and discoveries on viruses remain of lasting benefit to mankind.” Together, her remarkable contributions to structural studies in three separate areas, DNA, coal, and viruses, before her death at the age of 37 make her an inspiration to future generations of structural biologists, particularly women. We are proud to follow in her footsteps to use structural biology to gain insight into viruses with the goal of providing benefits to human health.

The severe acute respiratory syndrome coronavirus (SARS-CoV) epidemic of 2002, Middle East Respiratory Syndrome (MERS) epidemic of 2012, acquired immune deficiency syndrome (AIDS) pandemic starting in 1981, the Zika virus (ZIKV) epidemic of 2015-2016, and the ongoing SARS-CoV-2/COVID-19 pandemic are examples of the enormous global burden of viruses and the urgent need for vaccine and therapeutic development. Building on the prior contributions of early pioneers such as Rosalind Franklin, structural biologists continue to advance techniques in X-ray crystallography and cryo-electron microscopy (cryo-EM) to investigate viruses and viral proteins. We are interested in investigating antibody (Ab) recognition of viruses, which we do by solving 3D structures of viral proteins bound to Abs elicited by infection or vaccination. Understanding the structural correlates of Ab recognition of viruses is key for the development of effective monoclonal Ab therapies and vaccines (Figure 1).

Human immunodeficiency virus 1 (HIV-1) is responsible for the AIDS pandemic and 36 million deaths to date [6] and has long posed a challenge for vaccine development due its remarkable ability to evade the host immune response and establish latent reservoirs. HIV-1 contains a single viral protein on its surface that facilitates infection of immune cells. This protein, named Envelope or Env, is a trimer of gp120/gp41 heterodimers (Figure 2A). The gp120 portion of Env interacts with host CD4 receptors, which stimulates conformational changes that allow binding to the co-receptor, usually a host chemokine receptor called CCR5 [7]. These events trigger rearrangements in gp41 that allow fusion of the viral and host cell membranes, which is required for entry of the HIV-1 genome into the host cell [7]. In addition to small molecule anti-retroviral drug treatments to treat infected individuals, current strategies to prevent HIV-1 infection include vaccine design. Vaccine efforts seek to stimulate the evolution of broadly neutralizing Abs (bNAbs) that have been isolated in rare cases of human HIV-1 infection and are capable of broad and potent protection [8,9,10]. Advances in X-ray crystallography and cryo-EM have given us the invaluable opportunity to structurally characterize bNAb interactions with Env and Env conformational changes which have informed vaccine design efforts.

SARS-CoV-2, the virus responsible for the COVID-19 pandemic, has caused 4.5 million deaths and an estimated 225 million infections as of September 2021 [11]. The spike (S) proteins on the surface of SARS-CoV-2 allow it to infect host cells by binding the host cellular angiotensin-converting enzyme 2 (ACE2) receptor [12,13]. Each of the three protomers on an S protein includes two subunits, S1 and S2. The receptor binding domain (RBD) on S1 is the component that recognizes ACE2 during cell entry (Figure 2B) [13,14,15,16]. While the RBD can adopt both ‘up’ and ‘down’ conformations, it can only bind ACE2 when it is an ‘up’ conformation [14,15,16,17,18,19,20]. Due to the critical role of the RBD in facilitating infection, neutralizing Abs that target the RBD are an important component of the immune response against SARS-CoV-2 [21,22,23,24,25,26,27,28,29,30,31]. Structural biology has been instrumental in the rapid characterization and evaluation of the S protein and Abs produced in natural infection [15,21,22,23,24,25,26,27,28,29,30,31,32,33]. This work has contributed to the development of COVID-19 vaccines and monoclonal Ab (mAb) therapeutics, which have saved countless lives.

ZIKV is a mosquito-borne virus that can cause microcephaly and neurodevelopmental abnormalities in the newborns of infected mothers [34,35,36,37]. As part of the *flavivirus* genus, ZIKV shares similar features as other widespread flaviviruses such as dengue (DENV), West Nile virus (WNV), and yellow fever virus (YFV) [38,39,40,41]. Mature ZIKV has seven non-structural proteins (NS1, NS2A, NS2B, NS3, NS4A, NS4B and NS5) and three structural proteins: envelope (E), membrane (M), and capsid (C) [42,43,44]. The surface of ZIKV is coated by 180 copies of the E protein arranged as 90 dimers, and each E protein includes three ectodomains, EDI, EDII, and EDIII (Figure 2C) [38,42,43]. The flexible regions between the domains allows dynamic conformational changes to occur during viral entry and fusion [38,45,46,47,48]. EDII contains a conserved fusion loop (FL) peptide that becomes exposed after viral entry into cells and initiates endosomal fusion [42,43,49,50]. EDIII is thought to be important for receptor binding during infection, and consequently, is an important target for neutralizing Abs [51,52,53,54,55,56]. There is not yet a safe and effective vaccine against ZIKV that is universally available.

Here we review how approaches in structural and molecular biology have increased our understanding of Ab recognition of HIV-1, SARS-CoV-2, and ZIKV. We discuss how the design of stable and soluble viral antigens amenable for structural approaches has enabled our ability to analyze complexes of viral antigens bound by the antigen binding fragment (Fab) of Abs. Use of both cryo-EM and X-ray crystallography has increased our understanding of key viral epitopes targeted by Abs and conformational changes of viral proteins necessary for infection. These structural insights, combined with analyses of the levels of somatic hypermutation found in potently neutralizing Abs, provide valuable information for the development of effective vaccines and monoclonal Ab therapies to reduce global morbidity and mortality from epidemic/pandemic-causing viruses.

## 2. Main Body

### 2.1. Engineering Viral Surface Proteins for Structural Studies 

Structural biology techniques such as X-ray crystallography and single particle cryo-EM require samples that are stable enough to be isolated and manipulated in the laboratory. For some viruses, especially those that are symmetric, it is feasible to structurally characterize intact viruses using cryo-EM. For example, cryo-EM structures of intact, whole ZIKV have been solved with and without Fabs of Abs bound [42,43,57,58,59,60,61,62]. Viruses with pleomorphic structures (e.g., most enveloped viruses) can also be investigated structurally using cryo-electron tomography [63,64,65]. In order to prepare surface viral proteins of enveloped viruses for structural studies and therapeutic development, it has been necessary to produce soluble, native-like versions that are stabilized in a pre-fusion conformation that is targeted by neutralizing Abs.

Classically, the simplest way to solubilize a surface viral protein is to remove the transmembrane and cytoplasmic domains by truncation [66,67]. Truncation has also been used to produce smaller components such as single domains. Examples of truncated domains include gp120 cores of HIV-1 Env, which have the β4 and β26 strands and all flexible loops removed [68,69], coronavirus RBDs truncated at the base where the flexible hinge connects them to the rest of the S1 subunit [70,71], and the individual EDIII truncated from the rest of the ZIKV E protein [52,53,54,72,73]. Truncation of individual domains has been especially powerful for X-ray crystallography as crystallization is hindered by flexible regions such as loops or inter-domain linkers and hinges. Single domains are useful for solving high resolution structures of Fab-domain complexes that provide detail about the Ab interactions that may not be possible using single particle cryo-EM due to flexibility or heterogeneity of larger protein complex structures [32,52,53,54,72,73,74].

While truncated proteins have been useful in the field of structural biology, they do not necessarily reflect all aspects of the whole antigen and cannot always recapitulate the properties of a native viral protein. An extra layer of complexity exists since many viral proteins adopt distinct conformations depending on the step in the viral life cycle, requiring engineering and stabilization of the desired conformation for larger, multi-subunit complexes [43,46,47,57]. Fusion proteins such as Env and S include folded helical bundles that must extend for fusion of the viral and host cell membrane bilayers. These proteins are metastable in their pre-fusion conformation, which is usually the target of neutralizing Abs [75]. The introduction of stabilizing mutations can be helpful for preparing soluble constructs of larger, multi-subunit complexes. For example, helix-breaking proline mutations have been introduced into the central helices of fusion proteins, preventing the extension of helices required for membrane fusion [76]. In combination with an inter-subunit disulfide bond and truncation after residue 664, these mutations were introduced into HIV-1 Env to produce the pre-fusion stabilized SOSIP.664 trimers [77]. The proline helix-breaking stabilizing mutations have been successfully adapted to other viral fusion proteins including those on coronaviruses, RSV, Ebola virus, human metapneumovirus, and Lassa virus [75]. For SARS-CoV-2 S, additional prolines were introduced that further stabilize the trimer in the 6P, or ‘HexaPro’ version [67]. For studies of the ZIKV E protein soluble constructs of both monomeric E protein [56,78,79,80] and engineered disulfide-linked E protein dimers [55,81] have been designed.

Most regions of proteins have a purpose that is important to their function, particularly transmembrane regions and cytoplasmic tails [82]. Consequently, truncated and stabilized proteins used as substitutes for full-length equivalents are only useful to the extent that they are able to approximate the native state of the protein. It is essential for the engineered forms used for structural studies to be characterized with non-structural methods to confirm that they behave in a similar fashion to the native form in the context they are being studied.

### 2.2. Dominant Ab Epitopes on Viral Fusion Proteins

Structural analysis has facilitated identification of neutralizing epitopes on HIV-1, SARS-CoV-2 and ZIKV. Both X-ray crystallography and cryo-EM analyses of viral antigens in complex with neutralizing Ab Fabs have provided insights into mechanisms of neutralization by Abs and identified new therapeutic targets [7,32]. Neutralizing epitopes tend to be in structurally functional regions, and in many cases facilitate or hinder a structural change. In addition to neutralizing Abs, an immune response to a pathogen or vaccine can produce weakly neutralizing or non-neutralizing antibodies. which can be protective through various mechanisms such as antibody dependent cell cytotoxicity (ADCC) [83,84,85,86]. For viral fusion, there is typically a dramatic conformational change that occurs in the fusion protein to expose receptor binding sites for attachment and to insert the fusion machinery into the target membrane to undergo fusion [87]. Requiring a large conformational change is a strategy that allows viruses to hide vulnerable regions that are necessary for interactions important for viral function, such as target receptor binding. Many Abs bind in ways that can hinder or trigger fusion-necessitated conformational changes, resulting in various neutralization mechanisms [7].

#### 2.2.1. HIV-1 Env Epitopes

HIV-1 Env is present on the surfaces of virions in a closed pre-fusion conformation that includes centrally located gp120 subunits and the V1/V2 and V3 variable loops interacting about the apex of the trimer, hiding the co-receptor binding site on V3 [88]. Upon binding to the host cell receptor CD4 at the CD4 binding site (CD4bs) in the gp120 subunit, the Env protein rearranges to an open state in which the gp120s are rotated outwards, the V1/V2 loop is displaced to the sides of the Env trimer, and the V3 loop is exposed, allowing access to the co-receptor binding site on V3 [89,90,91,92,93] (Figure 3A). In the CD4-bound open conformation, a 4-stranded antiparallel bridging sheet is formed by the gp120 β-strands β20, β21, β2, and β3, the gp120 subunits swing away from the central axis and rotate slightly counter-clockwise, and the gp41 HR1 helices become more ordered and extended [90,91,92]. In this conformation, the V3 loop is exposed and can then bind to the co-receptor, which is required for entry [93]. HIV-1 Env epitopes target some of these intermediate fusion conformations, in addition to the closed, pre-fusion structure.

The epitopes of bNAbs often include conserved functional regions that are conformationally masked in the closed, pre-fusion structure or sterically restricted by N-linked glycans [7]. In fact, in many cases, N-glycans that occlude the protein surface of Env actually become part of the Ab epitope. HIV-1 epitope targets of bNAbs can be divided into the following categories: (1) bNAbs that bind at the apex of the trimer, specifically to the V1/V2 loops that undergo a dramatic rearrangement during host receptor engagement [94,95,96], (2) bNAbs against the V3-glycan patch, which includes the highly conserved GDIR motif and several N-linked glycans on and around the V3 loop [97,98], (3) CD4bs bNAbs that target the host receptor binding domain [7,74,99], (4) bNAbs that only bind to Envs in a CD4-induced open state [68,89,91], (5) “silent face” bNAbs that target a glycan-rich patch on the opposite face from the CD4bs on gp120 [98,100,101], (6) bNAbs that target the gp120/gp41 interface, including those that interact with the fusion peptide [102,103], and (7) bNAbs that bind to the membrane proximal external region (MPER) on gp41 [104] (Figure 4A).

Each epitope presents a distinct landscape for bNAb binding and poses different challenges for Abs to overcome. For most epitopes, N-linked glycans on the heavily-glycosylated Env trimer sterically restrict access to conserved protein regions, and therefore bNAbs tend to include conserved N-linked glycans in the epitope and/or develop long complementary determining region (CDR) loops to penetrate through the glycan shield [7]. This is the case for V1/V2, V3, and silent face epitopes. For example, the V3-glycan patch epitope is defined by the V3 loop that is essential for co-receptor binding and several N-linked glycans. bNAbs that target this region, including 10-1074, PGT121, and BG18, have long, 20+ amino acid CDRH3 loops that reach through the glycan patch to bind a conserved V3 motif from gp120 residues 324-327 with the sequence GDIR [97]. These bNAbs also make important contacts with conserved glycans Asn156_gp120_ and Asn332_gp120_. In contrast, some bNAbs against the CD4bs require short CDR loops to accommodate an N-linked glycan in that region. CD4bs bNAb 3BNC117 has a 5-residue deletion in CDRL1 that is necessary to prevent steric clashes with the Asn276_gp120_ glycan and a short, 5-amino acid CDRL3 that is essential to avoid clashes with gp120 [105]. The gp120-gp41 interface epitope is composed of protein and glycan residues in both subunits. This category includes bNAbs that target the fusion peptide (FP), which are the highly conserved N terminal residues of gp41 responsible for burying into the host cell membrane during the fusion process of viral entry. FP bNAb VRC34.01 binds primarily to the N-terminal 8 residues of gp41 with the remainder of interactions made with Asn88_gp120_ [102]. Together, these examples demonstrate the diverse epitope landscape of the HIV-1 Env trimer and how Abs develop particular features to overcome challenges posed by the dense glycan shield.

The mode of binding for bNAbs at all epitopes has been greatly illuminated by structural biology. In particular, X-ray crystallographic and cryo-EM structures of Ab:Env complexes have been essential tools to characterize which epitope newly isolated bNAbs bind, the mode of binding implemented, and to understand the context of atypical features in the sequence such as CDR lengths. The wealth of structural data has enabled structure-based design of gp120 and SOSIP-based immunogens that seek to elicit responses to particular epitopes and design small molecule drugs. 

#### 2.2.2. SARS-CoV-2 S Epitopes

The SARS-CoV-2 fusion machinery is the surface protein S, which is composed of three identical subunits each containing an RBD that sits at the apex of S and is attached to the rest of the subunits with a flexible hinge [14]. The RBDs are able to sample a ‘down’ conformation that hides the ACE2 binding site by packing it against a neighboring RBD, or an ‘up’ conformation, which exposes the ACE2 binding site at the tip of the RBD and is required for host receptor binding [15,32,106] (Figure 3B). 

Abs that recognize the RBD of the SARS-CoV-2 S protein are a vital part of the neutralizing Ab response to infection and vaccination because the RBD contains the binding site for ACE2. Effective neutralization by many anti-RBD Abs is due to their ability to block the RBD from binding the host ACE2 receptor. The epitopes targeted by Abs against the RBD can be organized into four simplified classes [32]. Class 1, *VH3-53/VH3-63*-derived Abs, target epitopes overlapping with the ACE2 binding site and only bind ‘up’ conformation RBDs. Class 2 Abs target epitopes overlapping with the ACE2 binding site and can bind both ‘up’ and ‘down’ RBDs. Class 3 Abs target epitopes that do not overlap with the ACE2 binding site and bind both ‘up’ and ‘down’ RBDs. Finally, class 4 Abs target a cryptic surface facing the S trimer interior and only bind ‘up’ RBDs [32] (Figure 4B). 

While the anti-SARS-CoV-2 Ab landscape has primarily focused on the RBD, a growing number of neutralizing Abs that target other regions of the S protein are being found. Neutralizing Abs that bind to the N terminal domain (NTD) [107,108,109] and the S2 domain [107,110,111,112,113] have been reported, indicating that the RBD is not the only site of neutralization. In addition, some of these Abs are also broadly cross-reactive to other betacoronaviruses as they target highly conserved regions of S such as the class 4 cryptic epitope on the RBD [24,114,115,116] or the stem helix of S2 [111,112,113]. 

#### 2.2.3. ZIKV Epitopes

The E protein of ZIKV and other flaviviruses is key for facilitating cellular entry and fusion [48]. The mature structure of ZIKV displays smooth virus particles with 180 copies of the E protein arranged as 90 dimers with icosahedral symmetry, and EDIII is thought to be responsible for binding cellular receptors [43,48,117,118,119,120]. After cellular entry through receptor-mediated endocytosis, the acidic pH triggers a conformational change by which the E proteins form trimers and expose the FL on EDII for membrane fusion [121,122,123].

Given its role in fusion, the E protein is an important target of neutralizing Abs that effectively clear ZIKV, inhibit ZIKV infection in vitro, decrease vertical transmission, and are protective in ZIKV challenge in animal models [53,54,55,56,72,79,124,125] (Figure 2C). Structural characterization of Abs that bind the ZIKV E protein have revealed multiple epitopes on the three domains: (1) the conserved FL found on EDII [56,126], (2) EDIII [52,53,54,72,73,127], (3) multiple domains of single E protein [79,80], (4) multiple domains spanning an E protein dimer [55,60,79,125,128,129], and (5) multiple domains spanning neighboring E dimer pairs [61,62,79,129]. Abs against the FL in EDII compose a large portion of the response to infection, and because the FL is conserved among flaviviruses, these Abs can cross-react with different flaviviruses [56,79,124,125,126,130]. However, many potently-neutralizing Abs target EDIII and these Abs tend to be more specific for ZIKV than other flaviviruses [51,52,53,54,55,56,61,72,124,125,131,132,133,134,135,136,137] (Figure 2C).

Notably, some Ab epitopes characterized by crystallography are not accessible on the known cryo-EM structures of mature ZIKV [43,54,56,126] (Figure 3C). While cryo-EM structures show a static envelope, evidence suggests the E proteins are dynamic and sample different conformations. The phenomenon of flavivirus “breathing” may result from conformational changes of the E protein during the viral life cycle, such as during fusion. The flavivirus DENV serotype 2 (DENV2) structure showed E protein rearrangements when heated to 37 °C, providing further evidence for flavivirus breathing [47,138] (Figure 3C). However, ZIKV maintains a smooth structure at 40 °C and its breathing conformation has not yet been determined [42].

### 2.3. Somatic Hypermutation of Neutralizing Abs

Abs evolve to neutralize antigen targets through the process of affinity maturation. This process begins when germline-encoded B cell receptors interact with an antigen and receive signals from T cells. This activation stimulates iterative rounds of somatic hypermutation (SHM), whereby a cellular mechanism orchestrates single base pair mutations, insertions, and deletions (indels) primarily in the CDRs of Abs [139]. These mutations are random, although favorable mutations that enhance recognition of antigen are selected for in further rounds of SHM [139]. Affinity maturation can rapidly diversify the Ab repertoire, allowing for the recognition of innumerable antigens that can mutate to evade Ab recognition [140]. This arms race between distinct Abs and antigens has been monitored through structural biology, which can illuminate how SHM impacts the antigen:Ab interface. For different viral antigens, SHM plays different roles in overcoming infection.

In HIV-1 infection, SHM plays a major role in the creation of bNAbs. Human Abs that have undergone affinity maturation on average carry 15–20 nucleotide mutations in the variable heavy (V_H_) gene; however, HIV-1 bNAbs include 40-100 V_H_ gene mutations [141]. High levels of bNAb SHM are necessary to combat a rapidly evolving antigen target in which Env mutations are selected to evade bNAb recognition. In fact, these mutations have been deemed critical for recognition and neutralization of native viral envelopes, as unmutated germline precursors of bNAbs do not usually interact with viral Envs [142]. X-ray crystallography and cryo-EM have allowed for the characterization of bNAb SHM to understand how mutated residues interact with HIV-1 Env and confer broadly neutralizing activity and potency [7,143,144]. Structures of Env:bNAb complexes have identified individual SHMs that are critical for neutralization activity at different epitopes and have set forth criteria for predicting the capability of newly isolated bNAbs.

Furthermore, structural biology has given context to unusual bNAb characteristics brought on by SHM; namely, framework region (FWR) mutations and indels. The FWRs of an Ab variable domain are the relatively constant sequences that provide a scaffold for the more diverse CDR loops. SHMs in FWRs are often poorly tolerated as they impair the structural integrity of the Ab [145,146,147]. However, HIV-1 bNAbs FWR SHM has been found to be critical for breadth and potency [145]. Analysis of crystal structures of bNAbs bound to gp120s revealed that regions of FWR SHM can directly interact with the antigen to increase the binding affinity or contribute to the structural rigidity and flexibility of a Fab for optimal binding [99,145]. HIV-1 bNAbs also contain unusually high levels of SHM indels [73,76]. Prior studies reporting sequences of Ab genes from memory B cells found between 1–3% of Ab genes contained indels [148]. For HIV-1 bNAbs, approximately 40% of bNAbs include indel mutations that range from 3–33 nucleotides in length [73,76]. Analysis of crystal structures of bNAb:gp120 complexes found that these indels are preferentially found within 10Å of the Ab:antigen interface [149]. Indels are therefore important to optimize interactions with Env, specifically to penetrate the dense glycan shield. Thus, structural biology has aided in elucidating how unusual SHM features in HIV-1 bNAbs contribute to breadth and potency.

Unlike HIV-1 bNAbs, Abs against SARS-CoV-2 S and ZIKV E protein have much lower levels of SHM and, in fact, affinity maturation via SHM is not always required to interact with their viral antigen targets [23,54,150]. Longitudinal studies tracking Ab evolution after SARS-CoV-2 infection found 1.3 months post-infection averages of 4.2 V_H_ and 2.8 V_L_ nucleotide mutations [151]. However, after 12 months past infection, SHM increased to approximately 15 V_H_ and 8 V_L_ nucleotide mutations [152]. Low levels of SHM have also been reported in longitudinal studies tracking ZIKV infection and comparisons of mature and germline versions of anti-ZIKV Abs [52,54,56,150,153,154]. Inferred germline Abs have been shown to be able to bind and even weakly neutralize ZIKV [52,54,153,155]. Structural analysis of Ab:antigen complexes for SARS-CoV-2 and ZIKV suggests most SHMs are found in CDR loops and contribute to the complex interface to create optimal contacts for antigen recognition [25,32,52,54]. For both of these viruses, the relatively low levels of SHM indicate near-germline and germline Abs are readily capable of recognizing viral antigens and maturing into potently neutralizing Abs. 

### 2.4. Structure-Guided Design of Vaccines, Small Molecules Inhibitors, and Ab Therapeutics

Structural biology has played a pivotal role in characterizing the optimal human Ab response which vaccines and therapeutics can be designed to mimic (Figure 1). For many viruses, the ability to produce a cross reactive response either to many strains of the same virus or to different viruses in the same group is necessary for complete protection from disease, presenting a challenge for vaccine design [156]. Therefore, the structure-guided development of small molecules, peptides, and protein decoys as therapeutics is a complementary strategy for treating viral infection [157]. 

Structural biology has allowed for the advancement of structure-based vaccine design, which is considered to be one of the current avenues most likely to eventually lead to an HIV-1 vaccine after the failure of subunit vaccines [158]. bNAbs are only elicited by a small subset of the population infected with HIV-1; even so these ‘elite controllers’ still never clear the virus [159]. In fact, arguably the biggest hurdle in creating an HIV-1 vaccine is eliciting an immune response that is far better than what is observed in infected people. The vast number of HIV-1 strains means a vaccine must protect against initial infection of countless distinct viral species rather than a single, or only a few, strains. Due to the inherent difficulty of eliciting bNAbs against HIV-1, some current structure-based efforts for HIV-1 vaccine design rely on structurally characterizing bNAbs in an effort to reverse engineer an immunogen that can elicit them, rather than the commonly-observed strain-specific, autologous neutralizing responses [158]. Structures of antigen:Ab complexes have allowed for the classification of Abs by their epitopes, which is necessary for the design of effective therapeutic monoclonal Ab cocktails [7]. In many cases, the dosing of single monoclonals is often suboptimal due to the ability of viruses to rapidly mutate. For example, in the case of HIV-1, the viral swarm inside a patient can evolve resistance mutations that make a therapy either less or not effective within days to weeks [160]. Therapeutics have also been designed to mimic an existing interaction by binding directly, such as CD4 mimetic drugs that bind into the CD4 pocket on gp120 [161,162]. As an alternative therapeutic approach, binding targets separate from canonical interaction sites can be used to inhibit function by preventing conformational changes, such as inhibitors directed at the HIV-1 Env fusion peptide [163].

In the case of SARS-CoV-2, which mutates at a lower frequency than HIV-1 but whose variants of concern are posing current problems, future efforts will need to focus on producing vaccines that are effective in the face of new variants [164,165]. Key regions of the S protein are highly conserved across the subgenus of sarbecovirus coronaviruses, of which at least three others can infect human cells: SARS-CoV, SHC014, and RaTG13 [166]. Neutralizing Abs that target the S of SARS-CoV-2 and also bind and neutralize other sarbecoviruses including SARS-CoV-2 variants of concern have been identified by several groups and have been structurally characterized [24,112,113,114,115,116,167,168], suggesting that an immunogen could be designed to produce a pan-sarbecovirus vaccine. Additionally, therapeutic mAb cocktails have successfully been developed for the treatment of SARS-CoV-2 [169]. Therapeutics have also been designed to mimic an existing interaction by binding directly, such as ACE2-S small protein decoys [170].

For ZIKV, design of a safe vaccine is complicated due to the similarities in structures between ZIKV and other flaviviruses. Since the structure of ZIKV is similar to that of DENV, WENV and YFV [38,39,40,41], there is concern that Abs elicited during infection with one flavivirus may cross-react with, but not neutralize, other flaviviruses during a later infection. This cross-reactive Ab recognition may worsen symptoms due to a phenomenon termed Ab-dependent enhancement (ADE), by which Ab-bound viruses can infect cells through interactions of the Fc regions of the bound Abs with the host Fcγ receptor, resulting in infection of cells after endocytosis of the Ab-virus complex [38,130,154,171,172,173,174,175,176,177,178]. This is of particular concern for the mosquito-borne virus DENV, since it has been shown that prior DENV or ZIKV infection that results in low or intermediate Ab titers increases the risk of worsened disease severity from a subsequent DENV infection with a different serotype [38,179,180,181,182,183,184,185,186,187]. However, potent neutralizing Abs against ZIKV EDIII have been identified that appear to be more specific for ZIKV than other flaviviruses, suggesting ZIKV EDIII is a potential candidate for the design of a safe vaccine [51,52,53,54,55,56,61,72,124,125,131,132,133,134,135,136]. No vaccine is yet universally available for ZIKV, although both the full E protein and individual EDIII have been investigated as potential immunogens [78,135,188,189,190,191,192,193,194].

## 3. Conclusions

Structural biology has allowed for a deeper understanding of the immune responses to many viruses, including HIV-1, SARS-CoV-2, and ZIKV discussed here. Mutations have been engineered that stabilize surface proteins in their pre-fusion conformations for use as starting immunogens for structure-based vaccine design and as laboratory reagents that can be used to study other aspects of the elicited humoral immune response. Structures of Abs bound to these stabilized proteins have allowed for the elucidation of neutralizing epitopes on the viral surface proteins. Additionally, such structures have increased our understanding of the role of features that Abs develop in response to antigens, such as somatic hypermutation, insertions, and deletions. For targets where whole inactivated or subunit vaccines have failed, structures of viral antigens bound to elicited Abs have facilitated alternative routes for structure-based design of vaccines, small molecules therapeutics, and Ab cocktails. It is through structural biology, inspired by advancements by Rosalind Franklin, that we are able to make progress toward vaccines and Ab treatments for the viruses we study, including HIV-1, SARS-CoV-2, and ZIKV.

## 4. Methods

Biorender.com was used to produce portions of Figure 1 and Figure 2. All structure renderings were made using PyMOL ver. 2.5.0. or 1.7.6.4.

Figure 4 was produced in PyMOL by aligning the HIV-1 Env or SARS-CoV-2 S proteins of each Fab-bound structure with the structure of the viral protein depicted in the figure (Env PDB: 5T3Z, S PDB: 7K8V). Only V_H_V_L_ domains are shown for each Ab. 

## Figures and Tables

**Figure 1 viruses-13-02106-f001:**
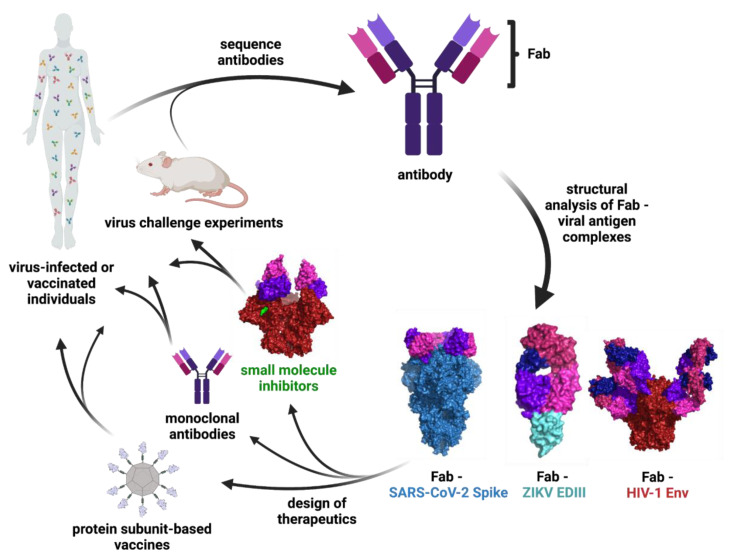
Schematic of Ab characterization and therapeutic development. The binding epitopes of Abs isolated from infected or vaccinated individuals or animal studies are determined through structural analysis of Fab—viral antigen complexes. These structures inform the design of vaccines, monoclonal Abs, and small molecule therapeutics that can be tested in clinical trials and animal models. Surface representations are shown for the following structures: Fab—SARS-CoV-2 S (PDB 7K90), Fab—ZIKV EDIII (PDB 5VIG), Fab—HIV-1 Env (PDB 5T3Z), and small molecule inhibitor—HIV-1 Env (PDB 7LO6).

**Figure 2 viruses-13-02106-f002:**
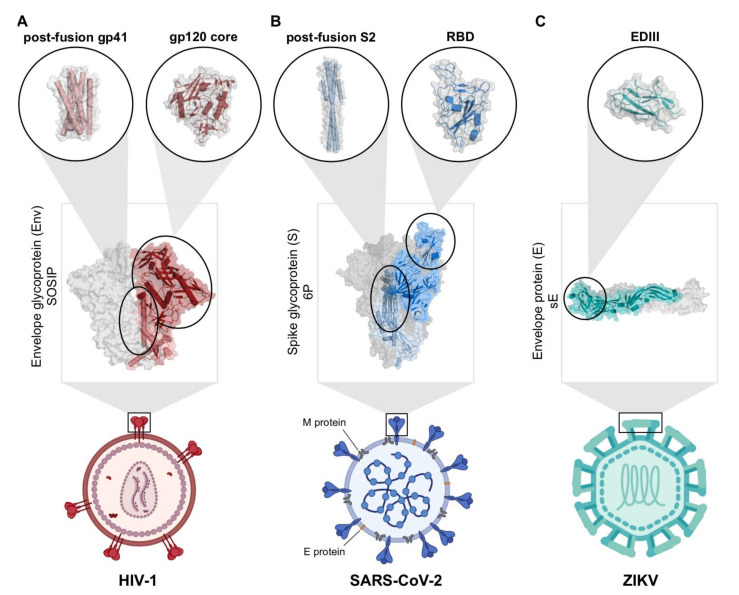
Structural targets of HIV-1, SARS-CoV-2, and ZIKV. (**A**) Cartoon HIV-1 virion with a closed, pre-fusion Env protein structure highlighted (PDB: 6UDJ). Circles show crystal structures of the postfusion gp41 bundle (left, PDB: 1AIK) and gp120 core (right, PDB: 5F4P). (**B**) Cartoon SARS-CoV-2 virion with S protein (blue), M protein (grey), and E protein (orange). The closed, pre-fusion S protein structure with one ‘up’ RBD (blue subunit) and two ‘down’ RBDs (grey subunits) is shown in the box (PDB: 7K8V). Circles show postfusion S2 helices (left, PDB: 6LXT) and RBD (right, PDB: 7K8M) structures. (**C**) Cartoon ZIKV virion with E protein (teal). The soluble E (sE) protein dimer structure is shown in the box with one E protein highlighted (PDB: 5JHM). The EDIII structure is shown in the circle (PDB: 6UTA).

**Figure 3 viruses-13-02106-f003:**
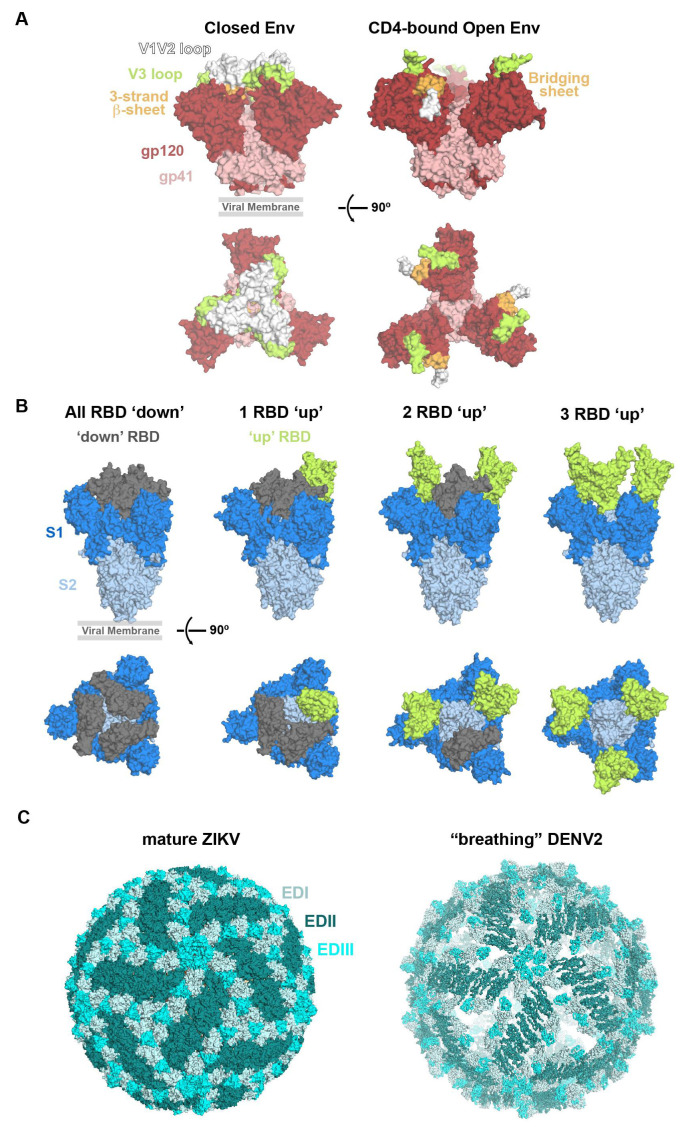
Conformational changes of HIV-1 Env, SARS-CoV-2 S, and ZIKV E. (**A**) Surface depictions of top down and side views of (left) closed, pre-fusion Env (PDB: 6UDJ) and (right) CD4-bound open conformation Env (PDB: 5VN3) highlighting the V1V2 loop (white), V3 loop (green), and 3-strand beta sheet (bright orange). Gp120 = dark red, gp41 = salmon. (**B**) Surface depictions of side and top down views of closed, pre-fusion S with three ‘down’ RBDs (grey, PDB: 7K90), 1 ‘up’ RBD (green, PDB: 7K8V), 2 ‘up’ RBDs (7K8Y), and 3 ‘up’ RBDs (6XCN). The location of the viral membrane is indicated in side views of viral proteins. (**C**) Surface depictions comparing the smooth mature ZIKV (PDB: 6CO8) and spiky “breathing” DENV2 (PDB: 3ZKO) structures. In the “breathing” DENV2 structure, EDI and EDIII of the E protein are protruding, giving the virus a “spiky” appearance, and holes are found in the surface.

**Figure 4 viruses-13-02106-f004:**
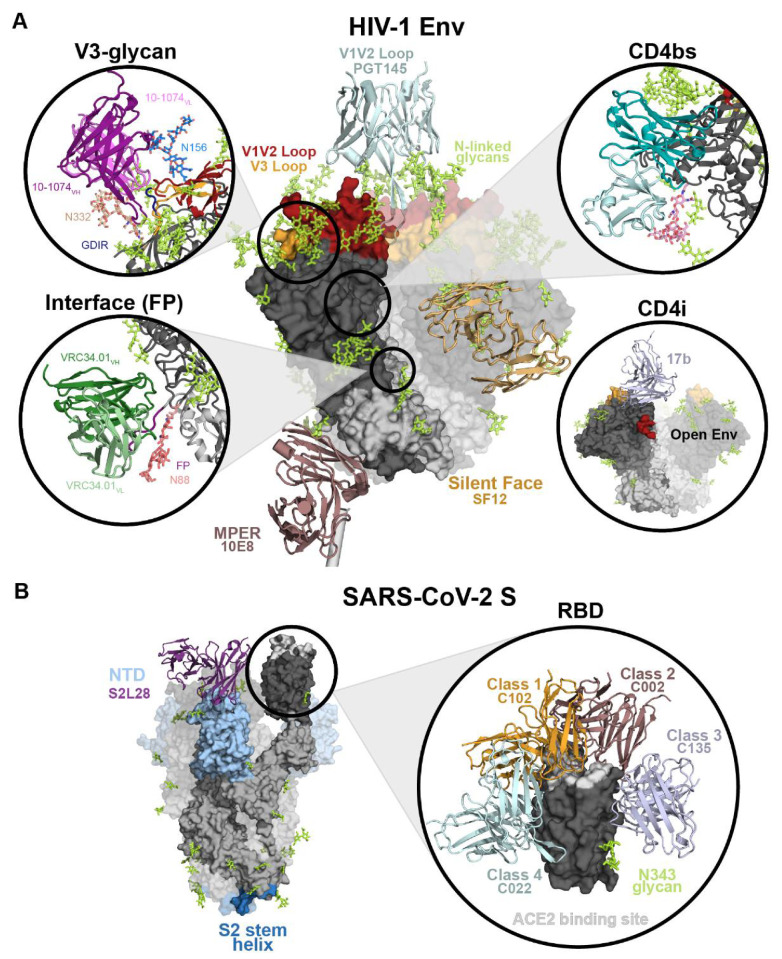
Neutralizing epitopes on HIV-1 Env and SARS-CoV-2 S. (**A**) HIV-1 Env structure (left) highlighting epitopes of representative bNAbs for each bNAb class. Env (PDB: 5T3Z) is shown as a surface with green N-linked glycans shown as sticks. Gp41 is light grey and gp120 is dark grey except for the V1V2 loop (dark red) and V3 loop (light orange). V_H_V_L_ domains of Abs binding the epitopes MPER (mauve, 10E8, PDB: 6VPX), V1V2 loop (pale cyan, PGT145, PDB: 5V8L) or Silent Face (sand, SF12, PDB: 6OKP) are shown as cartoons. Circles show details for Ab binding to the V3-glycan (10-1074, PDB: 5T3Z), interface (VRC34.01, PDB: 5I8H), CD4bs (3BNC117, PDB: 5V8L), and CD4i (17b, PDB: 7LO6). (**B**) SARS-CoV-2 S protein structure (left) highlighting the RBD (dark grey), S2 (blue), and NTD (light blue) Ab binding regions. V_H_V_L_ domains of Ab binding to an NTD (S2L28, PDB: 7LXX) epitope is shown as a cartoon representation. The circle (right) shows an enlarged view of the RBD surface with V_H_V_L_ domains for RBD-binding Abs shown as cartoons: Class 1 (light orange, C102, PDB: 7K8M), Class 2 (mauve, C002, PDB: 7K8S), Class 3 (pale purple, C135, PDB: 7K8Z), and Class 4 (pale cyan, C022, PDB: 7RKU). The ACE2 binding site is highlighted on the RBD in white.

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
