# Peer review of "How Antibodies Recognize Pathogenic Viruses: Structural Correlates of Antibody Neutralization of HIV-1, SARS-CoV-2, and Zika"

_viruses, 2021, doi:10.3390/v13102106_

Round 1
Reviewer 1 Report
The manuscript entitled, How antibodies recognize pathogenic viruses: Structural correlates of antibody neutralization of HIV-1, SARS-CoV-2, and Zika, uses simple and clear language to describe several antibody features against important immunogenic proteins and domains of viruses of public health concern from different families. This review is easy to comprehend and cite relevant literature and will be of great interest for a broader audience. It summarizes several important epitopes within HIV, SARS-CoV-2, and Zika virus that can be used for therapy and/or vaccine design. This article should be recommended for publication.
Author Response
October 8, 2021
Dear Mr. Vujinović,
Thank you for the insights on our resubmitted manuscript (ID viruses-1404618). We provide a point-by-point response to address reviewers’ comments below.
Reviewer Comments:
Reviewer 1
The manuscript entitled, How antibodies recognize pathogenic viruses: Structural correlates of antibody neutralization of HIV-1, SARS-CoV-2, and Zika, uses simple and clear language to describe several antibody features against important immunogenic proteins and domains of viruses of public health concern from different families. This review is easy to comprehend and cite relevant literature and will be of great interest for a broader audience. It summarizes several important epitopes within HIV, SARS-CoV-2, and Zika virus that can be used for therapy and/or vaccine design. This article should be recommended for publication.
We thank the reviewer for their appreciation of our manuscript.
Reviewer 2 Report
Please refer to the comments in the attached PDF

Author Response
October 8, 2021
Dear Mr. Vujinović,
Thank you for the insights on our resubmitted manuscript (ID viruses-1404618). We provide a point-by-point response to address reviewers’ comments below.
Reviewer Comments:
Reviewer 2
My compliments to the authors for writing a very well laid out review. The authors have considered a very relevant and broad topic of antigen-antibody interactions from several perspectives and mainly focused on the structural work done in the field. The review is free of proofreading and grammatical errors. The authors have also considered everything from somatic hypermutations to how different variants of the same virus might escape otherwise broadly neutralizing antibodies. However, there are a few changes and minor revisions that I would like to recommend before the review is accepted for publication.
Comments –
- This is perhaps a matter of creative differences between how I see the structural work being done on virus-antigen reactions vs how the authors describe it. The authors have mostly considered broadly neutralizing antibodies. Extensive work from the Saphire lab (LJI, La Jolla, CA) and Alter Lab (Harvard, MA) has come up with fascinating insights on virus-antibody reactions. Current data points to neutralization as a screen to select "good" antibodies whether for diagnostics or therapeutics while straightforward, is a bit limited. Most of this work relates to Filoviruses and Arenaviruses while some recent work relates to COV2. Adding a panel of tests that also detects protection in addition to neutralization has resulted in an array of dozens if not hundreds of mAbs some of which are 1) neutralizing but not protective, 2) moderate neutralizers but highly protective 3) weak neutralizers but moderately protective. In addition "recognition" of virus proteins by the Fab region, the role of the Fc receptor attaching to other immune cells has been found critical in the entire saga of antigen-antibody reactions. I would recommend to slightly modify the author's views here. Only slightly... or maybe add a separate paragraph. It seems like not considering protective antibodies seems like not considering interesting biology of virus- antibody interactions.
We thank the reviewer for raising this important topic. Although this review focuses on the role of structural biology in furthering our understanding of neutralizing antibody interactions with viruses, we acknowledge that there are other aspects of antibody immunity that are also important. Fc gamma receptors are a key component of the immune response to infection, including the negative consequences of antibody-dependent enhancement for flaviviruses discussed in the manuscript as well as the protective mechanisms of antibody dependent cell cytotoxicity (ADCC) and other Fc-mediated effector functions. We have added text to the manuscript to include non-neutralizing antibodies and ADCC as follows with appropriate references:
“In addition to neutralizing Abs, an immune response to a pathogen or vaccine can produce weakly neutralizing or non-neutralizing antibodies. which can be protective through various mechanisms such as antibody dependent cell cytotoxicity (ADCC).”
- Line 135, Line 159 – Excellent points!
- Figures – Authors should better reference their figures in this manuscript. Figures 2A and 2B have not been referenced at all. Also, Figures typically should be referenced in numerical order - Figure 1B before Figure 2A.
We thank the reviewer for careful reading. We have included call-outs for Figures 2A-B. All figures are now appropriately referenced in numerical order.
- Section 3.2 - I would recommend definitely refer to work by Hastie et al. 2021, (https://www.science.org/doi/10.1126/science.abh2315) and include this comprehensive study in the CoV2 specific section. I am aware that the article came out much later than your submission of the article, but it would be beneficial to have the authors’ view on this relevant body of work.
We thank the reviewer for pointing out this citation, which is a valuable reference for the field. This paper is now cited in the SARS-CoV-2 section about variants.
- Why is there a Zotero embedded link in your list of citations? Whenever I click on a doi it sends me to a Zotero Google blank page. The reference 105 from your lab only lists some authors and then has et. al. Is this a system default or word limit? There could be more such instances, but I am using this as an example. I don’t think list of references is included in the word limit.
We thank the reviewer for surveying our references. The embedded links were a result of formatting changes and they have been removed. Currently, there are no embedded links, so readers will have to copy and paste the doi to access publications we have cited unless the editor sees fit to add appropriate links.
The Viruses journal recommends the MDPI citation style that was used in this review. According to MDPI rules, “For documents co-authored by a large number of persons (more than 10 authors), you can either cite all authors, or cite the first ten authors, then add a semicolon and add ‘et al.’ at the end” (https://mdpires.com/data/mdpi_references_guide_v5.pdf). Although references do not count towards the word limit, we would prefer to keep the citation style used as our citation manager automatically uses this option.
- Figure 3B - The Zika virion rendering is a bit unclear. How are the authors depicting on a cryo-EM structure a surface epitope was "hidden"? Authors may have to zoom in and use a virtual slice to depict that. In the DENV structure, what part indicates that this structure is "breathing"? Typically this is displayed by a before and after map/structure with slightly altered conformations. Are the white patches where the surface proteins sort of "unlock" locally to get into this second conformation? Unfortunately, a non-virology science audience will have no idea what to look for. Even a non-virology cryo-EM scientist may not intuitively know.
Thank you for this suggestion to more clearly describe breathing conformations in the manuscript. As the reviewer insightfully pointed out, depicting the phenomena of flavivirus “breathing” is a challenge because Zika likely samples multiple heterogenous conformations, and there is no “breathing” structure of Zika available for direct comparison to the mature structure. Additionally, the published breathing structure of DENV2 is limited in resolution (13.7 Å) (Fibriansah et al. (2013)). Due to the challenges in showing the details of the complex conformational changes, Figure 3C focuses on the overall conformational change from smooth (mature Zika) to spiky (breathing DENV2). Cryptic Ab epitopes are not portrayed because antibodies bind several different epitopes of the E protein (some partially hidden and some accessible in the mature Zika structure) and due to the challenges in clearly depicting these cryptic epitopes within the overall quaternary structure. To aid the audience in identifying conformational differences during breathing, we revised the manuscript as follows:
“The phenomenon of flavivirus “breathing” may result from conformational changes of the E protein during the viral life cycle, such as during fusion. The flavivirus DENV serotype 2 (DENV2) structure showed E protein rearrangements when heated to 37°C, providing further evidence for flavivirus breathing [48,139]. Compared with the smooth structure of mature ZIKV, the breathing DENV2 structure includes protruding EDI and EDIII domains, giving the virus a spiky appearance, as well as holes in the surface (Figure 3C). However, ZIKV maintains a smooth structure at 40°C and its breathing conformation has not yet been determined [43].”
We also revised the text in the Figure 3C legend:
“Surface depictions comparing the smooth mature ZIKV (PDB: 6CO8) and spiky “breathing” DENV2 (PDB: 3ZKO) structures. In the “breathing” DENV2 structure, EDI and EDIII of the E protein are protruding, giving the virus a “spiky” appearance, and holes are found in the surface.”
Additionally, we found a mistake in the coloring of Figure 3C, and a revised version has been added.
- In antibody research mentioned across this review, have any of the cited references talk about non neutralizing monoclonal antibodies?
Esswein et al. 2020 discusses how some weakly and non-neutralizing antibodies isolated from Zika-exposed individuals cross-react with, but do not neutralize other flaviviruses (e.g. West Nile Virus) and contribute to antibody-dependent enhancement through Fc interactions. In addition, we have added a citation about potential roles of non-neutralizing antibodies in immune responses to other viruses.
We thank the reviewers for the careful reading of the manuscript and excellent suggestions for improving our paper.
Sincerely,
Pamela J. Bjorkman